# The Importance of Being Local: The Role of Authenticity in the Concepts Offered by Non-Themed Domestic Restaurants in Switzerland

**Robert Home** [1],*[iD], **Bernadette Oehen** [1], **Anneli Käsmayr** [2], **Joerg Wiesel** [2] and **Nicolaj Van der Meulen** [2]

1   Department of Socioeconomics, Research Institute of Organic Agriculture (FiBL), 5070 Frick, Switzerland; bernadette.oehen@fibl.org

2   Hochschule für Gestaltung und Kunst, Fachhochschule Nordwestschweiz (FHNW), Münchenstein b., 4142 Basel, Switzerland; anneli.kaesmayr@fhnw.ch (A.K.); joerg.wiesel@fhnw.ch (J.W.); nicolaj.vandermeulen@fhnw.ch (N.V.d.M.)

*   Correspondence: robert.home@fibl.org

**Abstract:** In the highly-competitive restaurant environment, restaurateurs continually optimize the quality of their offer so that customers leave the restaurant with the intention to return and to tell others about their experience. Authenticity is among the attributes that restaurateurs seek to provide; and a wealth of study has been conducted to understand authenticity in a variety of contexts including ethnic-themed restaurants. However; insufficient attention has been given to non-themed domestic restaurants; which make up a significant proportion of available dining options. This study aimed to explore the role of authenticity as part of the concepts offered by domestic restaurants in Switzerland. Interviews with managers of 30 domestic restaurants were analyzed according to their content and interpreted according to authenticity dimensions identified by Karrebaek and Maegaard (2017) and Coupland and Coupland (2014). The approach of using a framework with four dimensions—"tradition", "place", "performance", and "material"—was a useful epistemological lens to view the construct of authenticity. Participants from country restaurants placed more importance on tradition, while restaurateurs from both country and urban restaurants emphasized the importance of seasonal and regional ingredients and of fitting the restaurant within the cultural and geographical landscape. Managers of domestic restaurants in Switzerland see authenticity as a way of attracting and retaining customers, which can thereby contribute to the economic sustainability of restaurants, although the participants cautioned that customer expectations of sufficient choice can outweigh the added value of authenticity. Authenticity of the product offered by domestic restaurants can also contribute to the sustainability of place by enabling continuity of cultural heritage and traditions. These results provide a basis for future research that could guide restaurateurs' decisions of how to include authenticity when developing and implementing their concepts for domestic restaurants.

**Keywords:** authenticity; domestic restaurants; tradition; place; performance dimension; material dimension

## 1. Introduction

In a world filled with deliberately and sensationally staged experiences, consumers increasingly base their purchase decisions on their perceptions of how authentic they perceive an offering to be [1]. The growth in consumer interest in the authenticity of products, services, performances, and places has been reflected by a corresponding interest by scholars [2]. A base assumption of study into authenticity is that its attribution conveys value to the object of study, which has been supported by research, such

as that of Kovacs et al. [3] who demonstrated that authenticity generates higher consumer value ratings of organizations. Carroll [2] similarly noted that consumers in modern society increasingly embrace products, services, and forms of expression that exude and exemplify the authentic. The reaction from industry has been to make claims of "real" and "authentic" on product packaging, but Gilmore and Pine [1] claim that authenticity is a construct of perception and that nothing from businesses is truly authentic. Furthermore, Coupland [4] raised questions about whether the concept of authenticity in experience is desirable in every context.

There appears to be a consensus that authenticity plays an important role in how people understand, and relate to, objects and experiences [5]. In the vast philosophical literature, scholars have proposed numerous dimensions of authenticity, depending on context, which have been broadly categorized into objective, constructive, and postmodern dimensions [6]. However, in the context of authenticity of experience, Belhassen et al. [7] challenge whether authenticity can be objectively determined and propose that that authenticity perception is related, negotiated, and dependent on context. Indeed, authenticity is usually described as a social construction rather than a sum of the specific characteristics of the products or services [8,9]. Furthermore, authenticity can be treated from a relational perspective, i.e., by comparing "the original" to "the copy" in terms of mimetic features, such as asking whether an interpretation is true to the original historical, social, or cultural context [10].

One of the two interpretations of authenticity identified by Carroll and Wheaton [11] is "type" authenticity, in which an entity is evaluated in regard to the degree that it is true to its associated type, category, or genre. It is not difficult to imagine how this might apply to types of restaurants. Indeed Skinner et al. [12], in the context of restaurants, take the view that authenticity is socially constructed and that evaluation of authenticity is based on individual subjective perceptions. However, Jugård and Modig [13] noted that the meaning of authenticity with regard to the food industry remained rather unexplored in academic literature. There seems to have been little progress towards an accepted definition of food and dining authenticity in the intervening decade. This paper takes the view that authenticity in the context of restaurants is both socially constructed [12] and relational [10], which implies that the perceived degree of authenticity of a restaurant is evaluated against some explicit or implicit criteria.

The authenticity of experiences, as a relational construct, means that it is built around the relationship between expectations and the perceived experience. Inherent in most understandings of authenticity is the dependence on the perceptions of the consumer, which implies that the expectation has been communicated, either directly or indirectly, by the supplier of the experience. Carroll [14] (p. 126), suggests that authenticity should be directly and explicitly communicated if impact is to be maximized, and that the communication of authenticity should be "tightly and visibly integrated into the structure of an organization". This sentiment is supported by Jugård and Modig [13] who wrote that authenticity is characterized by being true to what is communicated to others, so that personal interests and passion are perceived and that the consumers' common perceptions match the communicated expectations. An example of indirect communication is the "service-scape", including the physical setting, which Wang and Matilla [15] suggest provides cues that create pre-consumption expectations with regard to authenticity, and which are mediated, based on the consumer's previous experience and familiarity.

Researched into authenticity in sectors that have a relationship to the food service context has commonly found that it can add value to products and experiences, and thereby attract customers and encourage repeat custom. For example, Robinson and Clifford [16], in their study of foodscapes in a medieval festival context, found associations between perceived authenticity and re-visitation intentions. Beverland [17] noted that consumers of luxury wines perceive that authenticity adds value to the product for a range of reasons including linking to the past and indicating a consistent level of high quality. Derbaix and Derbaix [18], in their study of attendees at "generational" music concerts, found that perceived authenticity positively correlated with value ratings. Castéran and

Roederer [19] reported that authenticity added quality, and thereby value, to the experience of attending the Strasbourg Christmas market.

One context in which authenticity does appear to have become an institutionalized attribute is food and dining [2], and there appears to be little dispute that authenticity is relevant to restaurants [8]. In these days of unprecedented access to information and increased competitiveness, it has become crucial for restaurateurs to provide and communicate the factors that customers use for selecting their dining experience [20]. Jugård and Modig [13] wrote that restaurants must render authenticity, and thereby go beyond just offering food and common services, if they are to be a part of the booming experience industry. In the context of restaurants, Kovacs et al. [3] used an authenticity scale to find that consumers perceive independent, family-owned, and specialist (single-category) restaurants as more authentic than they do chain, non-family-owned, and generalist (multiple-category) restaurants. Furthermore, they concluded that authenticity generates higher consumer value ratings [3].

The majority of prior study into the perceptions and utility of authenticity in the restaurant context has focused on ethnic or themed restaurants, with results that suggest that restaurant customers typically perceive authenticity in objective and constructive ways [21]. Jang et al. [22] investigated Korean restaurants in the U.S. and found that tangible elements, such as décor, were more important for high end restaurants, while food related attributes were more important for casual-dining restaurants. Skinner et al. [12], in their study of tourists' experiences of visiting local restaurants in Greece, found that décor was less important than whether local people eat there and whether they serve local food and wine. Nichele [23] reported that authenticity was not regarded as essential in the experience of dining in lower-scale Italian restaurants in Lancaster, England, which is in contrast to Gilmore and Pine's [1] assertion that authenticity is what customers really want. Song et al. [24] reported that the ethic appearance of the waiting staff and other customers, along with the customer's knowledge of the ethnic cuisine, influence authenticity perceptions in Chinese restaurants. Common to these results is that authenticity is an interplay between the tangible elements provided and the expectations of the customer.

Although highly relevant, the focus of the vast majority of the literature regarding authenticity in food and dining being on ethnically themed restaurants [21] has missed a significant proportion of existing restaurants: Local domestic restaurants that are non-themed. In addition to being plentiful, domestic restaurants are also worthy of study because of their social role. Di Pietro and Levitt [21] point out that domestic restaurants perform a vital role in communities by serving as manifestations of a region's history, identity, and heritage, while the role of ethnically themed restaurants, such as Chinese restaurants outside China, could be described by Karrebaek and Maegaard [25] as recreational. Furthermore, domestic restaurants serve dishes that are consumed by a populace on a daily basis, and thus contribute to sense of place. Despite their number and their importance, there has been insufficient study to allow conclusions as to whether domestic restaurants also have the capacity to include the concept of authenticity in their offer.

The aim of this study is to elaborate on existing theories of authenticity to understand the role of authenticity in domestic restaurants by examining whether authenticity is perceived to add value to the dining experience. Examining authenticity in the context of domestic restaurants will help to separate the authenticity dimensions that are associated with creating an authentic dining experience outside the context of that experience (e.g., eating Chinese food outside China), from the dimensions that are associated with domestic dining experiences. These latter dimensions might be considered to be less context dependent and therefore more generalizable, and thereby provide a framework of studying authenticity of restaurants in a variety of contexts.

## 2. Methodology and Research Design

Given the wealth of study of authenticity in other contexts, the verbal deductive approach appears appropriate in that it builds on the concepts defined in a review of relevant literature to generate an explanatory theory, including concepts, assumptions, and implications, for a new context [26].

Therefore, this approach allows prior study to be used to develop a theory to understand the contribution of authenticity to the experience provided by domestic restaurants. We focus on the perspective of the providers of the experience: The restaurateurs, whose financial survival relies on meeting the needs of their customers. We argue that few people will study the expectations of customers with comparable interest, intensity, and duration.

## 2.1. Recruitment Procedure

We took a purposive sampling approach [27] to identify non-ethnically themed restaurants that aim to provide services for local customers. Restaurants were selected, by means of local knowledge supplemented by an internet search, to represent a wide range of qualities and regions, including the German and French speaking parts of Switzerland, with care taken to select restaurants from urban areas and the countryside. Regions of Switzerland with tourist based economies were specifically avoided in the sampling strategy, and restaurants that are actively directed towards providing services for tourists were excluded. All of the selected restaurants serve a mixture of traditional and modern dishes, which is typical in Swiss domestic restaurants. We chose three each of fine dining, mid-range, and low end restaurants in both urban and rural settings in the German speaking region of Switzerland. These 18 restaurants were supplemented with a fine dining, mid-range and budget restaurant in both urban and rural settings in the French speaking part of Switzerland and in the Emmental region, which is known for its adherence to Swiss traditions. This constellation resulted in a sample of 30 restaurants, which are shown in Table 1. These restaurants were approached with an initial telephone call to arrange a time to visit and conduct interviews.

**Table 1.** Typology of participating restaurants.

| | German Speaking | | French Speaking | | Emmental Region | |
|---|---|---|---|---|---|---|
| | **City** | **Country** | **City** | **Country** | **City** | **Country** |
| Fine dining | 1: O, L, N<br>2: O, B, S<br>3: C, B, N | 1: O, L, N<br>2: O, B, N<br>3: C, L, S | C, B, N | O, L, N | C, B, N | O, L, S |
| Mid-range | 1: O, B, N<br>2: O, L, S<br>3: C, L, N | 1: O, L, S<br>2: C, B, N<br>3: C, L, N | C, B, N | C, L, S | C, B, N | C, L, N |
| Low end | 1: O, L, S<br>2: C, B, N<br>3: C, B, N | 1: C, B, N<br>2: C, L, S<br>3: C, B, S | C, B, N | C, B, S | C, B, N | C, B, S |

**Legend:** O: Restaurant uses mostly organic ingredients; C: Restaurant uses mostly conventional ingredients; L: Most ingredients are sourced from local suppliers; B: Most ingredients are sourced from bulk handlers; S: Restaurant employs mostly local staff; N: Restaurant employs mostly non-local staff.

## 2.2. Data Collection

Qualitative interviews were conducted on site with representatives from the 30 restaurants in which participants were asked: What the concept behind their cooking is; why they chose that concept; how much freedom they have in driving the concept; how much of the concept comes from the customers; what experience they wish their customers to remember; how they communicate the experience; how their relationship with customers has changed over time; how they choose what to put on the menu; how they choose where to source their ingredients; what they consider the added value of local and/or organic; and whether they would do something differently if they were to move their restaurant to a different location. The interviews were recorded, transcribed, and translated into English.

### 2.3. Theoretical Framework

Newman and Smith [5] point out that the wealth of academic interest and research into the concept of authenticity has not yet resulted in a widely accepted definition of the term so several scholars have taken the promising approach of defining authenticity according to underlying dimensions. Zanchetti et al. [28] defined authenticity in terms of three major dimensions: The "material" dimension, which refers to tangible material aspects of authenticity, the "constructive" dimension, which refers to an interaction of intangible aspects with the material, and the "expressive" dimension, which has a great emphasis on the intangible aspect of authenticity. Coupland and Coupland [29] organized discourses within four frames, which they refer to as "material", "cultural", "recreational", and "performative", with each frame defining a social dimension in which authenticity can be experienced. Together, these four frames provide a practical framework for analyzing authenticity in the context of recreational experience.

The approach of using dimensions of authenticity has also been used with food and dining. Johnston and Baumann [30] wrote that "geographic specificity", which refers to the place of origin of the recipes, "naturalness", and "tradition" are prominent among the salient dimensions that frame the qualities of authenticity. Specifically referring to gastronomy, Karrebaek and Maegaard [25] identify "tradition" and "place of production" as the most important dimensions, although they noted that "cultural", "recreational", and "material" dimensions also play a role. Lego Muñoz and Wood [31] suggested that expectations are also mediated by distance from the host country, which can only apply to ethnically themed restaurants.

To provide a framework for analysis, we selected three authenticity dimensions: "tradition" "place", and "material" as suggested by Karrebaek and Maegaard [25], along with the "performative" dimension, as suggested by Coupland and Coupland [29]. We considered Coupland and Coupland's [29] "cultural" dimension, in the context of domestic restaurants, to be part of the "place" dimension, so did not consider it separately. Furthermore, Coupland and Coupland's [29] "recreational" dimension, as it is commonly understood, does not appear to be easily transferrable to dining experiences in domestic restaurants, so was not included as a dimension in this analysis.

### 2.4. Analysis

The interviews were analyzed according to their content [26] using the software package MaxQDA, and text was coded, with the four selected dimensions that provide the first order of coding. The interviews were coded by one researcher, with the coding scheme then discussed within the research team until agreement was reached on the final coding scheme. Throughout the presentation of the results of the analysis, direct citations from restaurateurs are shown in inverted commas

## 3. Results

### 3.1. Tradition Dimension

The words "regional" and "seasonal" were universally nominated as part of the restaurant concept by country restaurateurs, and most concepts also included some basis of connection to the land. Traditional dishes, which are part of the cultural landscape, is a strong influence in country restaurants and is similarly based on connection with the land: "People want potato dishes. Potatoes are at the heart of the Swiss food culture." Several of the sampled restaurants had been inherited from parents and grandparents, so the concept was also inherited, and there was no apparent reason to change it. In general though, country restaurateurs had not given a lot of thought to concept as connection to the land was considered to be self-evident. Part of the country feeling is that "traditional Swiss" is an attractive part of it and a common response was that this connection was expected by their clientele and was what they wanted to present. Tradition has two meanings in country restaurants. On one hand they tend to have a regional tradition (Swiss food), while on the other they can also have the tradition that is peculiar to the restaurant, such as a specialty they are known for in the

community. For example, one country restaurant is known for meringue and people travel to the country specifically for this specialty.

Urban restaurateurs, on the other hand appeared to have given more explicit consideration to their concept and the tradition dimension tends to refer to the traditions of the individual restaurant. Consequently, respondents from urban restaurants tended to report a need to find a defining attribute that is interesting and brings something new to the community. However, the tradition dimension remained relevant, even when the traditions are internal, because customers become used to a restaurant having some specialties and are disappointed if they change or become unavailable. Sometimes the reputation and the concept develop without a conscious decision by the restaurateur. For example, in a city restaurant, the restaurateur decided to offer burgers because they were "still in a tight price range and wanted just to make sure we didn't miss something. We wanted customers to have tastes and fragrances of everywhere". After a period of time, people would seek the restaurant for the burgers, so more were added to the range, and the restaurant became known locally as a burger restaurant.

Once an attribute becomes part of the restaurant's tradition, there is sometimes an expectation that this will remain constant. This finding is reflected in the statistics that are kept on the food sales, with restaurateurs universally aware of which items from their menus are the best sellers. One restaurateur referred to finding a gap, or niche, in the market and then communicating that promise, which was summarized as part of the identity of the business. This highlights the importance of the image, or reputation, of the restaurateur, which reflects the image of the restaurant. However, tradition, and a restaurant's history, also bring limitations to change, which some restaurateurs find to be restrictive. A city restaurateur commented: "we have to continue this concept, I mean, I have to keep going. If we stop the cordon bleus, we can close." A similar restriction is also perceived by country restaurateurs who report the necessity of having schnitzel with French fries on their menus. Limitations of tradition also have implications on the variety on offer. A common solution is to always have a part of the menu that is fixed, which is complemented by an extra menu in which the cooks can express themselves or a seasonal menu that can be experimental.

Common to both urban and country restaurants is the activity of adding authenticity by highlighting traditions of long term relationships with suppliers. Relationships with suppliers are perceived to contribute to authenticity and are openly communicated by describing the relationships on the menu in text and sometimes pictures. The origin of ingredients is often included in the titles of the dishes, which indicates the importance. One urban restaurateur insists that deliveries of produce are brought through the dining area of the restaurant, so that diners see them arriving, while a country restaurateur adopts a similar strategy in that the fresh produce is delivered to the reception at the time that the hotel guests are usually departing. In addition to adding to perceptions of authenticity, there are practical benefits to pursuing traditions in terms of the reliability that stem from long-term collaboration. One country restaurateur insisted: "This isn't a thought about margin, but about our region. We have to contribute. When we don't do anything, we can't wonder why there's no butcher anymore, no dairy, and no bakery and so on". Loyalty works both ways and small supply businesses have a clear interest in reliably delivering to the restaurants who are their regular customers.

## 3.2. Place Dimension

The choice of local ingredients can be seen to contribute to a sense of place because of the implicit connection with the landscape: A lakeside restaurant, for example, concentrates on fish dishes. In many cases, the menu is dictated to a large extent by what is produced. However, local produce is affected by concerns about reliability in that there is a lack of confidence in the supply. Depending on how "local" is defined, there might simply not be enough of a particular product grown in a local region to provide the quantity required by a commercial restaurant. Another restaurant buys organic produce from a neighboring village, and pays a premium for the produce, but does not mention the word "organic" on the menu because of fears that the supply might be insufficient for them to honor their promise. In that case, the organic produce is chosen because of its quality and because it is local. In urban restaurants,

organic ingredients appear to be given a higher profile in the menus, while less emphasis is given to the localness of ingredients. In country restaurants, the opposite appears to be the case, with the contribution of local to added value being more than that added by organic: "For me, it's much better that I can say that I get berries locally. It's much more valuable than organic berries from elsewhere".

A further aspect of place, in addition to connecting the customer to the place of production of the ingredients is related to the interplay between the location and surroundings of the restaurant with the other authenticity dimensions. In the case of restaurants, it appears reasonable that fitting within the landscape, as well as interpreting the landscape in the food that is offered, might contribute to authenticity. The relevance of linking with the countryside is different for city and country restaurants. The experience of eating in a country restaurant begins with the journey to the restaurant: "A lot of people say 'pity you're so far away', but that also has its attraction. The journey also belongs to the experience". Travel to that particular restaurant is through the countryside, and parking is in what is clearly a farmyard. Customers see farmland out of the window of their cars and start to connect with the countryside. In contrast, the experience of eating in urban restaurants tends to start at the door of the restaurant.

In addition to the connection to place that comes with local produce is the concept of seasonality, which also contributes to authenticity. Restaurateurs report that local people tend to know what is in season, so seeing those ingredients on a menu gives a feeling of having insider knowledge. The connection with the land that several of the responding restaurant concepts are based around would not work without the seasonality. Customers like to understand what is going on and both city and country restaurants have seasonal menu changes. A country restaurateur pointed out that: "For us it's that we change the menu four times per year. There are four seasons". In one urban restaurant, they take pride in being creative with ingredients that are commonly believed to restrict creativity, such as wild oats, and point out that providing seasonal produce is giving the customers what they want. This attitude might be seen as connecting the restaurant with the landscape where the food is produced: A challenge that is greater for urban restaurants than for country restaurants. However, and similarly to the restrictions associated with sourcing local produce, seasonality brings restrictions, which while contributing to authenticity, may be not accepted by customers.

Mobility has grown and restaurateurs are limited by their need to satisfy customers who expect to be able to eat what they want. Restaurants have to remain practical if they are to remain in business and authenticity can be given less importance than convenience. The practicality of the decision-making is summed up by a restaurateur who explained: "Customers like the idea of eating locally, but what interests them first and foremost is what's on their plate. Commercially, there are limits". A city restaurant admitted sourcing unseasonal leafy vegetables in winter that are grown in heated greenhouses in Holland. A common compromise is to include seasonal ingredients as special additions to a menu, but to maintain a "core" menu that remains constant: Even when the ingredients are out of season. In this way, the customer's demand for authenticity is satisfied without impacting on their choice.

### 3.3. Performative Dimension

Some restaurateurs explicitly include a reminder that food has an origin in the country within their restaurant concept: "I think the concept is not to forget where we come from, the inclusion of people, animals, and environment." This connection was highlighted in both urban and country restaurants by stage managing the delivery of fresh produce through the front door during mealtimes, in view of the customers, while dry goods and packaged ingredients are delivered through the back door. Although this performance is not a demonstration of authenticity in the strict sense, it contributes by reminding the customer of the connection with the local environment. A similar connection by performance was observed in an urban restaurant that has a garden near the entrance in which small quantities of the types of vegetables that are included on the menu are grown. Although this "sow

garden" cannot conceivable produce enough to supply the restaurant, it gives the illusion of a garden to plate concept.

Local customs in Swiss gastronomy are based around individual contact with customers. For example, it is normal to greet customers individually when they arrive and to say goodbye when they leave. Communication of the concept, and the consequent creation of expectations, takes a range of forms including personal communication, material presentation and, increasingly, social media. One restaurateur commented that experience is communicated on a personal level by an explicit strategy of establishing a friendly relationship with the customers, based around the idea of openness and honesty. However, personal relationships require personnel, who are expensive, so there are practical limitations on personal communication. Many restaurateurs express that it is worth the effort: "I think we run an honest business. We respect our clients. We respect our staff. I think that shows". In addition to local customs that are common to a region, customs can be specific to a particular restaurant and developed in a relatively short period of time. For example, a restaurant that has a tradition of large and freshly made schnitzels keeps the door open so that customers can hear the schnitzel being hammered.

Employees are one area in which many restaurants deviate from the principle of sourcing resources locally. In Switzerland, the wages and the standard of living are high while the wage paid to serving staff is comparatively low: "I'd like to have just Swiss, but the zeitgeist has gone so far that it's not possible anymore". Although this takes away from the authenticity, it has become sufficiently accepted that it is not perceived as being a significant loss. Another restaurant employs solely local people, but on a casual basis, with the employees typically working one day a week. The restaurateur feels that the organizational difficulties of working with multiple people are compensated by the added authenticity given by local staff who speak the local dialect and have a good idea about what's going on locally.

*3.4. Material Dimension*

Everything that is presented contributes to an expectation and "putting the concept together is quite complex because it needs to work sustainably for a long time so you have to think about every factor of the business". For example, one restaurant embodies the traditions of the city in which it is located, with decorations that recall the traditional architectural style of the last century: Old tiled stove, wooden furniture (table, chairs and banquets), wood-clad walls with carved faces on the beams that watch over the table for the regular customers. A country restaurant is in a converted typical farmhouse with parquet floors and farm-kitchen style furnishings. In a lakeside restaurant, paintings of local fish are prominent and embody the concept of the restaurant, and are there to promote the reputation of the local fish, while simultaneously connecting with place. The dining room in another restaurant is decorated in Swiss farmhouse style with the message that the diners are on a farm to eat farm produce. Common to both urban and country restaurants is that menus are only in the local language (German or French) although some restaurants also have an English language version available on demand.

Fundamental to any restaurant, and the primary means of communicating experience, is the quality of the food. Although thought goes into the details including the kinds of plates, how the food is organized on the plate, the utensils, and the pepper and salt shakers, the main focus is naturally placed on the ingredients and the recipes: "Food and drinks have to be good. When it's not good, sooner or later, you're out of business". The contribution of good food to authenticity is based around an implicit assumption in the minds of customers that authentic local food tastes good. Dishes have to be prepared to a higher standard than can usually be found in home kitchens, or with variations that amateur cooks might not think of. In the case of domestic restaurants, the requirement for the food to be of good quality is particularly high because the customers are familiar with the dishes and will have eaten variations of the same dishes elsewhere.

## 4. Discussion

### 4.1. Limitations

A limitation of this study is the sampling strategy in which only restaurateurs were included as interview respondents, so there is no evidence that authenticity, as perceived by restaurateurs (owners/managers), is consistent with that of customers who are the final arbiters of whether or not a restaurant's attempt to convey authentic elements is indeed perceived as adding value. Future research will be required to examine the nature of authenticity in relation to restaurants further. However, Coupland [4] raised questions about the feasibility of evaluating authenticity of experience, which leads to the question of whether the dimension-based understanding of authenticity might be readily operationalized in quantitative study. Nonetheless, the results of this study, which allow the hypotheses that authenticity can indeed contribute to the experience perceived by the customers and that authenticity can be readily understood in terms of dimensions, suggest the value of such an endeavor. Using quantitative approaches with larger representative samples to gather the perceptions of restaurant customers would contribute to addressing this limitation, while also contributing to the body of knowledge on the role of authenticity in providing positive restaurant experiences. A further limitation is that the qualitative sample was, by nature, small and was focused on Swiss restaurants. Although these limitations are common in qualitative study, they mean that the results are not necessarily generalizable, so further confirmatory research in other contexts would also be valuable.

Newman and Smith [5] pointed out that the wealth of academic interest and research into the concept of authenticity has not yet resulted in a widely accepted definition of the term. Despite this handicap, the dimension based framework, with dimensions of "tradition" "place", and "material" as suggested by Karrebaek and Maegaard [25], and "performative", as suggested by Coupland and Coupland [29], proved to be useful for examining authenticity in the context of domestic restaurants in Switzerland. The results suggest that restaurateurs see attributes that contribute to authenticity as an integral part of the experience they provide, and see them as contributing to customers leaving the restaurant with the intention to return and the intention to tell others of their positive experience. Lalicic and Weismayer [32] touched upon the notion of the relationship between authenticity and the quality of the perceived experience by considering authenticity as a dimension of experience. However, interpreting the responses to the questions about the development and delivery of the dining experience supports the value of considering authenticity as a multi-dimensional construct [28] that can be viewed through a number of epistemological lenses. The dimensions of authenticity that were revealed in this study are shown graphically in Figure 1 and will be discussed individually.

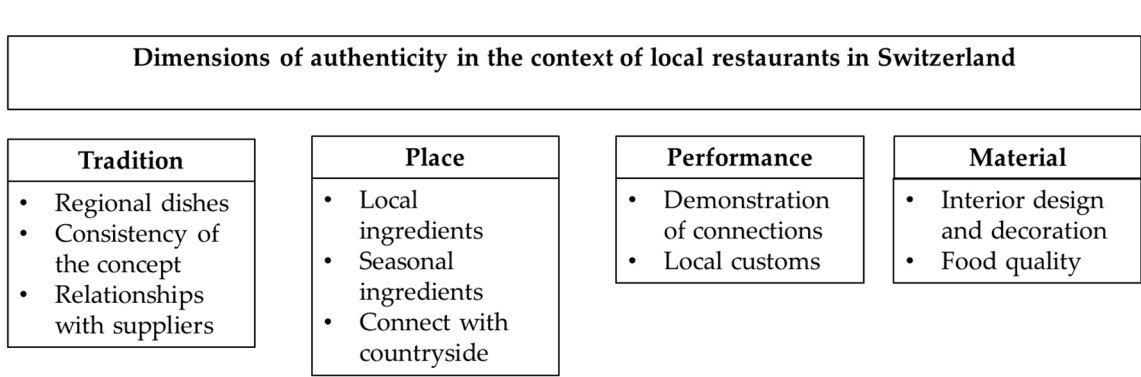

**Figure 1.** Dimensions of authenticity in the context of local restaurants in Switzerland.

### 4.2. Tradition Dimension

Intangible elements of identity and cultural landscape are commonly used to differentiate from competitors in the restaurant sector [33] and tradition was found to be an important concept to be communicated to customers in Swiss restaurants. This finding is consistent with the results of

Kim et al. [34] who proposed that history is antecedent to perceived authenticity. Karrebaek and Maegaard's [25] tradition dimension attributes authenticity to a phenomenon by connecting it to a time and space, and by recalling history, which is typically communicated by restaurants offering regional dishes. A further means of demonstrating tradition is by the material dimension, such as traditional décor or using traditional descriptors as text in the menus. Country restaurateurs in Switzerland especially perceived an expectation to provide more general Swiss traditions, including traditional menu items. Although country restaurateurs perceive the requirement to provide tradition to their customers as being restrictive, the strength of the demand means that maintaining local traditions is central to country restaurants and few restaurants were found to deviate from this perceived expectation.

Cities are much more eclectic and can be seen to be a celebration of diversity [35], so the scope for interpretation of what might be acceptable, or desired, by customers was larger. In the case of urban restaurants, the traditions tended to be specific traditions attached to the history of the particular restaurant, which contribute to the gastronomic niche it occupies. For example, a restaurant that is known locally for its tradition of making large and freshly made schnitzel has created a dimension of authenticity with its specialty. Once a restaurateur has established the gastronomic niche, the tradition dimension of authenticity suggests that they should remain consistent with the concept, and this point was consistently repeated by the participating restaurateurs.

Country restaurants can, of course, also maintain traditions that are specific to the particular restaurant, and indeed many do. However, these restaurant-specific traditions are typically in addition to the more general regional or national traditions they perceive are essential to their existence. Skinner et al. [12] reported that tourists place importance on freshly prepared local specialty dishes as an indicator of authenticity, so would consider most Swiss country restaurants to be authentic. However, local traditions specific to a particular restaurant might not be recognized by tourists because they are dependent on local knowledge. It appears therefore that these restaurant-specific traditions are of primarily of importance to local customers.

A further intangible element of the tradition dimension is the relationships with local suppliers. Carroll [2] illustrates the importance of authenticity to a domestic restaurant using the example of the famous Chez Panisse in Berkeley California. Although the restaurant was themed (French), Carroll's [2] description focused on fresh local high-quality, and increasingly organic, food items from a network of local suppliers. This description marks a departure of the discourse on authenticity from a focus on the authenticity of its theme (in this case it being authentically French) to a focus on the culture in which the restaurant was situated. The authenticity was expressed by the restaurateur being true to the local cultural landscape, including the local suppliers. The results of this study are consistent with these findings, with most participating restaurateurs placing a high degree of importance on maintaining relationships with local suppliers, which brings mutual benefits.

*4.3. Place Dimension*

In contrast to the "tradition" dimension, there were few fundamental differences between city and country restaurants in attitudes towards local and seasonal ingredients, with the primary difference being that city restaurants tended to highlight the connection to the countryside, while country restaurants tended to include the relationship to the ingredients as part of their identity. Beer [33] reported that society seeks to regulate the authenticity of food by using terms, such as local seasonal, and organic, that are consistent with the discourse of authenticity in dining experiences, The results of this study suggest that Beer's [33] findings might be applicable across a broad spectrum of restaurants including those that do not have an explicit theme. Lang and Lemmerer [36] point out the significant and growing influence of locally sourced and produced foods and ingredients on consumer food attitudes and behaviors, which has the potential to exceed the added value given by organic [27].

However, there are different interpretations of what constitutes local. For one participating restaurateur, local means being grown on site, while another demonstrates the localness of the

ingredients by cultivating a vegetable garden in the restaurant grounds so that diners can see what the ingredients look like on the plant. At the other end of the spectrum, a restaurateur considered "local" to mean choosing ingredients that could potentially grow in the region although the actual ingredients used may well have been grown elsewhere, while for another, "local" means local suppliers but not necessarily local produce. These findings shows a perception by restaurateurs that local ingredients indeed add value to the customers' experience, but also suggest that customers are prepared to accept that produce is local without seeking assurance. Indeed, restaurateurs who do source locally grown seasonal ingredients appear to concur with Boys and Blank [37] and tend to highlight this: Either materially, such as by writing details on the menus, or by performance, such as orchestrating deliveries of produce through the dining area.

Both urban and country restaurateurs in Switzerland seek to make connections with the food they serve and the countryside where the ingredients are produced. This finding is consistent with the conclusions of Karrebaek and Maegaard [25] who nominate place of production as one of the main dimensions of authenticity. In the case of restaurants, production, in the normal sense of the word, primarily takes place on site, so we understand this dimension to mean the place where the ingredients are produced. Indeed, this was a recurring theme in the interviews. In country restaurants with traditional Swiss décor, the "place" authenticity dimension is enhanced by, and therefore shares a large overlap with the "material" dimension of authenticity, which is in agreement with Zanchetti et al.'s [28] "constructive" dimension that refers to an interaction of intangible aspects with the material: In this case the intangible sense of place with the material experience of the decoration and how that fits within the landscape. Connection with landscape is not as obvious in an urban environment in that urban landscapes are man-made and often heterogeneous. However, it appears the concept of connection with place is relevant on the larger scale of the city being situated within a landscape in which food is produced. The responding restaurateurs appear to agree with Boys and Blank's [37] conclusion that local produce, and a connection with where food is produced, is an important influence on food choices such as those made by restaurant customers.

Implicit in the concept of local ingredients and a connection to the place of production of the ingredients is seasonality, which was commonly mentioned as a key concept: Particularly in the high-end and mid-range restaurants. Lang and Lemmerer [36] similarly identified the importance of seasonal foods with limited availability as a motivation for food preferences and classified this item under "authenticity benefits" in their study. However, seasonality also has disadvantages because some ingredients are sometimes not available, and customers may demand out of season food. In this way, the authenticity dimension of "place" is related to the "tradition" dimension and the restrictions that it implies. The finding that restaurants retain the need to be practical expands on the results of Lehman et al. [38] in that, while customers may be willing to forego hygiene for the sake of authenticity, they are less willing to forego choice, which maintains its importance: Even when authenticity would involve restrictions based on conformity to cultural norms.

Restaurateurs have developed several strategies to reconcile the conflicting goals of providing authenticity with local and seasonal ingredients and maintaining customers by providing sufficient choice. Among these is to provide a core menu that is independent of season, and to supplement that with a seasonal menu that satisfies the authenticity needs of the customers. A further strategy of gaining relief from the restrictions of product seasonality is to relax the understanding of what constitutes "local", which can lead to a wider range of local foods; a finding that supports the results of Lang and Lemmerer [36]. However, both of these strategies are based on the assumption that the customers will not critically reflect on the offer. Again, the restaurants that embrace the concept of seasonality, which are commonly the high-end and mid-range restaurants, tend to highlight it in the material dimension by adding text to the menus, such as including the word "seasonal" in the names of the dishes. This finding is similar to those of Carroll [2], with similar use of the concept of seasonality found during the beginning of the locavore movement.

### 4.4. Performative Dimension

The dimension of performative authenticity is concerned with the notion that authenticity in the tourist industry, such as medieval festivals, is based on selected elements, and sometimes on practiced and even scripted representations [16,25]. The premise of authenticity as a social construction in which "the original" is compared to "the copy" [8,9] is not readily transferrable to domestic restaurants because restaurants are objects of consumption, and as such part of an "entertainment package" [25]. Customer satisfaction is dependent on the restaurant being able to deliver the promised experience rather than by comparison with an original ideal.

The results of this study support the contention by Di Pietro and Levitt [21] that restaurants should promote their community connections and ensure their food and beverage meets customers' authenticity standards. Several restaurants demonstrate their community connections by arranging for local produce to be delivered through the dining area in full view of the customers, which is clearly performative. In this way, they use performance to showcase the relationships with suppliers that are part of the tradition dimension as well as their connection to the countryside, which is part of the place dimension. A further connection that can be performed is a demonstration that meals are prepared from raw ingredients. For example, the restaurant that has a tradition of large and freshly made schnitzels keeps the kitchen door open with the explicit intent of ensuring that customers can hear the schnitzel being hammered. This example illustrates how restauranteurs can also use performance to highlight their own traditions that they have developed over time.

Skinner et al. [12] found that an important indicator to tourists of authenticity is when local people eat at the restaurant. While this appears to be a performative aspect, it is not clearly relevant to domestic restaurants that aim to serve local people, such as in this study. However, a further characteristic in which performance is relevant to authenticity in domestic restaurants is by employing local staff who have a common background, including speaking the same dialect, and therefore some connection with the customers. Karrebaek and Maegaard [25] reached a similar conclusion when they highlighted the importance of employing staff who spoke the dialect of their Bornholm themed restaurant. The economic situation in Switzerland, with few local people prepared to work for the relatively low wages of service staff, has however made this an unrealistic proposition; especially for city restaurants and remote country restaurants in which the staff must live on site. Some country restaurateurs still however see the added value in employing local staff and continue to do so, but the contribution to authenticity is perceived to be waning over time.

### 4.5. Material Dimension

Karrebaek and Maegaard's [25] material dimension refers to the tangible characteristics perceived by the customer: from the large-scale design, such as the building and décor, through to small details such as the plates on which the food is presented. The result that interior design and restaurant décor is a cue to conveying local authenticity for local residents is in contrast to the preferences of tourists who Skinner et al. [12] reported placing low importance on these material attributes. The finding in this study that interior design contributes to the intangible feeling of tradition in country restaurants is in agreement with the findings of Karrebaek and Maegaard [25] who, in their study of constructions of authenticity, reported that décor, photos, tableware, narratives, and dialect features contribute to the construction of a comprehensive semiotic experience of a Danish, but not necessarily local, restaurant. Skinner et al. [12], in their study of domestic restaurants in a tourist setting, found that tourists perceived interior design and restaurant décor to be less important for domestic restaurants than for ethnic-themed restaurants, which contrasts the perceptions of Swiss restaurateurs who place value on the interior design being in harmony with the cultural and geographical landscape. The importance given by the responding restaurateurs to the message conveyed by the interior is however consistent with the concept of the service-scape proposed by Wang and Matilla [15]. A further tangible element that contributes to authenticity is when menus are written in the local language [12]. This criterion has

less relevance in the context of domestic restaurants in Switzerland because they all have menus in the local language although some may have some examples in a non-local Swiss language or English.

The quality of the food is clearly a material aspect of the quality of the experience and the link to authenticity: In particular, to "tradition" and "place", is clear. Indeed Lang and Lemmerer [36] included "better crafted foods" as an "authenticity benefit" to motivate preference for local foods. The promises are usually implicit and the importance of the material dimension does not appear to be consistent. Lower end urban restaurants were still subject to the perceived requirement to provide traditional local dishes, but less attention is needed for the material surroundings in the restaurant. Expression of both local and restaurant specific traditions by providing appropriate tangible characteristics, such as décor or food quality, was found to be more important for higher end restaurants and for country restaurants. This might be explained by customers who are paying what they consider to be a high price for a meal, or who have travelled a long distance to the restaurant, having invested more in their dining experience and might have correspondingly higher expectations. The imperative to deliver is greater when the investment made by the customer is greater. For example, in an expensive restaurant, or in a restaurant where the customer has to travel to get there, the customer "bets" their reputation by recommending the restaurant, so the consequences will be greater if it fails to deliver. This is in contrast to the expectations of restaurants for which the customer has invested less and therefore has correspondingly lower expectations [23].

## 5. Conclusions

There appears to be close to consensus among restaurateurs that attributes that contribute to authenticity add value to the experience of domestic restaurants in Switzerland, which supports Gilmore and Pine's [1] assertion that authenticity is indeed what customers want. Breaking down the responses about perceived expectations of authenticity in domestic restaurants, we can understand authenticity as the case in which the experience that is delivered to the customer matches that which was perceived to have been promised. The results could be classified against four main dimensions: Tradition, which is expressed in terms of serving regional dishes, consistency of the concept over time, and long term relationships with local suppliers; Place, which is expressed by serving local and seasonal ingredients, and making connections with the countryside where the food is produced; Performance, which commonly involves demonstration of connections and local customs; and Material, which describes tangible characteristics such as interior design and food quality. On a practical level, these results provide a basis for future research that could guide restaurateurs' decisions when developing and implementing their restaurant concepts.

Although Lego Muñoz and Wood [31] wrote that geographic location mediates perceptions of authenticity, they were specifically referring to the distance of an ethnically themed restaurant from the country of origin. This study also found that geographic location is an important mediator in the case of domestic restaurants, but for different reasons. No differences were found between restaurants from the different language regions, but some differences were found between urban and country restaurants. For restaurants located in the countryside, the experience of the restaurant, and therefore perceptions of authenticity, include the journey to the restaurant. The experience with city restaurants, in the case of Swiss cities, begins at the entrance of the restaurant. Wang and Mattila's [15] (p. 346) assertion that "authenticity assessment starts at the pre-purchase stage" therefore appears particularly applicable to country restaurants. The material dimension, which includes the interior design, the décor, and how that fits with the local environment, is particularly important for local people visiting country restaurants, which suggests that local people place a different value on these attributes than tourists who place low importance on these material attributes [12].

Although some differences were found between urban and country restaurateurs in terms of how they approach authenticity, many of the results were consistent across the respondents, including those from different language regions and different standards. These results suggest that consideration of authenticity can contribute to attracting and retaining customers and thereby

the economic sustainability of restaurants. Furthermore, it appears that domestic restaurants in Switzerland can contribute to sustainability of place by maintaining cultural heritage and traditions such as with family restaurants being handed down through generations and with the maintenance of food traditions. Although domestic restaurants exist primarily for local people, the overlap between the characteristics that contribute to their authenticity and the characteristics that tourists seek in domestic restaurants [12] suggests that domestic restaurants might also contribute to the tourism attraction of a region.

In any case, there is little doubt that authenticity is a crucial concept to restaurateurs and owners of domestic restaurants. Indeed, Lehman et al. [38] wrote that consumers form value judgments about restaurants based on a code of hygiene, which is rational and considers compliance with local health regulations, but which recedes in importance upon activation of a code of authenticity that involves conformity to cultural norms. In other words, customers may be prepared to forgo characteristics that are normally considered essential, such as hygiene, for the sake of authenticity [38]. Despite authenticity adding value, there is a balance between that benefit and the practicality, or cost, of being authentic. For example, although a local country restaurant might be more authentic if all the staff were also local, it is sometimes not possible to find affordable local staff. A further example is that, although local and seasonal produce contribute to being perceived as authentically local, this value is trumped by the expectation of the customer to have a sufficient choice.

Restaurateurs constantly weigh the benefits and costs of every decision to remain competitive in the restaurant sector. Many of the dimensions of authenticity could be gleaned from the responses to questions about their concepts and the opinions of the restaurateurs, with collectively many years of experience, report that maintaining tradition and providing connection to place are important to them. Furthermore, they report that demonstrating tradition and place with performance and tangible characteristics to fit the restaurant within the cultural and geographical landscape contribute to providing authenticity of experience. Although participants from country restaurants placed more importance on tradition, while restaurateurs from both country and urban restaurants emphasized the importance of seasonal and regional ingredients, the general consensus is that the added value of authenticity is worth the associated costs provided it does not excessively restrict the choices offered to the customers.

**Author Contributions:** The individual contributions of each co-author are specified as follows: Conceptualization, B.O., A.K., J.W., N.V.d.M. and R.H.; Methodology, B.O., A.K., J.W., N.V.d.M. and R.H.; Validation, B.O., A.K., J.W., N.V.d.M and R.H..; Formal analysis, B.O. and R.H.; Investigation, B.O., A.K. and R.H.; Resources, J.W. and N.V.d.M.; Data curation, R.H.; Writing—original draft preparation, R.H.; Writing—review and editing, B.O. and N.V.d.M.; Supervision, B.O., J.W. and N.V.d.M.; Project administration, J.W. and N.V.d.M.; Funding acquisition, J.W. and N.V.d.M. All authors have read and agreed to the published version of the manuscript.

**Funding:** This research was funded by the Swiss National Science Foundation within the research project: "Cooking and Eating as aesthetic Practice. An explorative survey", grant number 162891. The APC was funded by the Swiss National Science Foundation.

**Acknowledgments:** We sincerely thank all of the restaurant managers who took time from their always hectic schedules to answer our questions.

**Conflicts of Interest:** The authors declare no conflict of interest.

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
