# Peer review of "The Importance of Being Local: The Role of Authenticity in the Concepts Offered by Non-Themed Domestic Restaurants in Switzerland"

_sustainability, doi:10.3390/su12093907_

Round 1
Reviewer 1 Report
There is definitely a need for more research into non-themed domestic restaurants, therefore the subject matter alone requires consideration for publication.
The aspect of authenticity with regard to such restaurants is also a topic worthy of investigation, and one that should throw up interesting and original insights.
However, I am not a fan of single place-based cases in any tourism and hospitality research unless the results are clearly shown to evidence some generalisable contribution to practice, or the method employed show some contribution to theoretical development of the subject matter.
Methodologically, if the paper is claiming to consider the role of authenticity in experiences, and the manuscript notes that experiences are customer focused, and that “the growth in consumer interest in the authenticity of products, services, performances, and places” it seems at odds with the title and focus of the paper upon initial reading of the abstract for the data to be gathered from restaurant managers and not diners themselves.
It was heartening to see the research covering restaurants in both urban and rural settings.
There are many more authenticity dimensions and approaches to authenticity than applied in the method of data collection and analysis in this paper (see for example Skinner, H., Chatzopoulou, E. and Gorton, M. (2020) ‘Perceptions of localness and authenticity regarding restaurant choice in tourism settings’ Journal of Travel and Tourism Marketing, 37(2), pp. 155-168 who describe 8 types of authenticity *sorry if that seems a bit too “reviewer 2”, but the paper really has more relevance to this work than you have included in your manuscript*.)
Given that the issues of authenticity / localness / seasonality / organic ingredients arose from the data, it would have been a good idea to have included some more literature on some of these issues, or to consider more about the differences / similarities / implications of these concepts in the discussion and conclusion. For example, you could have made much more about the very interesting finding about urban and countryside restaurants that showed:
“In contrast to the ‘tradition’ dimension, there were few fundamental differences between city and country restaurants in attitudes towards local and seasonal, with the primary difference being that city restaurants tended to highlight the connection to the countryside, while country restaurants tended to include the relationship to the ingredients as part of their identity.”
I really think much more could be made about the implications of all of your findings in the analysis / discussion and conclusions section of the manuscript. You are saying what you found, but not necessarily what are the implications of your findings, both for scholars and for practitioners.
Another issue that has not been explicitly addressed is that of what type of customers the restaurants serve – are they local people or tourists? If tourists, are they international or domestic tourists? You state that “For restaurants located in the countryside, the experience of the restaurant, and therefore perceptions of authenticity, include the journey to the restaurant”, yet it would be interesting to know if this would be as true for local diners as for visitors from either Swiss cities or from other countries. Therefore, within your method section it would be interesting (if you already have the data, or if you could gather anything you would need to fill in the gaps in such a typology with a quick phone call to the restaurants) to include a table showing the typology of the restaurants sampled - including, for example, if they are in a city or the countryside, how long they have been established, the type of food they serve (e.g. traditional / modern, Swiss / international), if they use organic ingredients, if they use local ingredients, if they employ local serving staff etc
Given that you have submitted this manuscript in consideration for publication in the journal Sustainability, I would have expected to have seen at least some reference to what these findings mean for restaurant or place sustainability – you have already alluded to this with regard to the carrying on of tradition, family restaurants being handed down through generations, sustainability of cultural heritage and traditions (even if it is schnitzel with fries) etc. But all of these could have been mentioned more explicitly with reference to sustainability.
You conclude that “On a practical level, these results provide a basis for future research that could guide restauranteurs’ decisions when developing and implementing their restaurant concepts.” It would be useful to include a short summary paragraph of these results, showing what could / should be done for both countryside and urban restaurants to improve authenticity as something that can add value to diner experiences – especially as this reflects the title and focus of your manuscript.
In conclusion, I did enjoy reading this paper. It focuses on an interesting and relevant topic that has been under-researched in the extant literature. It therefore has good potential to add interesting and original insights into the subject area. However, I think it could be improved by considering some of the recommendations above, particularly:
- covering more of the concepts to be discussed in more depth in the literature review
- being more explicit about acknowledging the limitations of your methodological approach
- being much more explicit about the implications of the various findings in the analysis / discussion and conclusion because at the moment they tend to be more descriptive
Author Response
COMMENT: Methodologically, if the paper is claiming to consider the role of authenticity in experiences, and the manuscript notes that experiences are customer-focused, and that “the growth in consumer interest in the authenticity of products, services, performances, and places” it seems at odds with the title and focus of the paper upon the initial reading of the abstract for the data to be gathered from restaurant managers and not diners themselves.
RESPONSE: We didn’t intend to mislead. We have now changed the title to “The importance of being local: the role of authenticity in the concepts offered by non-themed domestic restaurants in Switzerland”, and have made some adjustments to the abstract to reflect this.
COMMENT: There are many more authenticity dimensions and approaches to authenticity than applied in the method of data collection and analysis in this paper (see for example Skinner, H., Chatzopoulou, E. and Gorton, M. (2020) ‘Perceptions of localness and authenticity regarding restaurant choice in tourism settings’ Journal of Travel and Tourism Marketing, 37(2), pp. 155-168 who describe 8 types of authenticity *sorry if that seems a bit too “reviewer 2”, but the paper really has more relevance to this work than you have included in your manuscript*.)
RESPONSE: The redesign of the structure has meant that further authenticity dimensions are mentioned in the introduction and the contribution of Skinner et al. is more strongly considered in the discussion.
COMMENT: Another issue that has not been explicitly addressed is that of what type of customers the restaurants serve – are they local people or tourists? If tourists, are they international or domestic tourists?
RESPONSE: We looked for restaurants that provide services primarily for local people. This has now been explained in the sub-section labeled ‘recruitment procedure’.
COMMENT: Therefore, within your method section it would be interesting (if you already have the data, or if you could gather anything you would need to fill in the gaps in such a typology with a quick phone call to the restaurants) to include a table showing the typology of the restaurants sampled - including, for example, if they are in a city or the countryside, how long they have been established, the type of food they serve (e.g. traditional / modern, Swiss / international), if they use organic ingredients, if they use local ingredients, if they employ local serving staff etc.
RESPONSE: A table has been added. This was not a trivial task as we hadn’t collated this data during our initial data collection. Most restaurateurs found it very difficult to answer how long they have been established. Several were uncertain whether to answer how long since they took over management, how long the restaurant had been a restaurant (in several cases over 100 years), or how long since their last change in direction, such as becoming an organic restaurant. For this reason, we left that data out of the table. It was also difficult to differentiate according to the other categories. For example, restaurants that specialize in using local ingredients also use some ingredients from elsewhere, so we chose the option to say ‘mostly local ingredients’. Similarly with organic. Several restaurants use organic ingredients but not exclusively, so we also opted for ‘mostly organic’ Finally, with staff, we took the same option and classified restaurants as staffed by mostly Locals or mostly non-locals.
COMMENT: It was heartening to see the research covering restaurants in both urban and rural settings.
RESPONSE: Thank you. We felt this was important in the Swiss context.
COMMENT: You state that “For restaurants located in the countryside, the experience of the restaurant, and therefore perceptions of authenticity, include the journey to the restaurant”, yet it would be interesting to know if this would be as true for local diners as for visitors from either Swiss cities or from other countries.
RESPONSE: We don’t have the data to answer this question. We do intend to follow this study with a survey of diners in Swiss restaurants and will be sure to collect data to answer this question then.
COMMENT: being more explicit about acknowledging the limitations of your methodological approach
RESPONSE: There is now a limitations paragraph at the beginning of the discussion section.
COMMENT: However, I am not a fan of single place-based cases in any tourism and hospitality research unless the results are clearly shown to evidence some generalisable contribution to practice, or the method employed show some contribution to theoretical development of the subject matter.
RESPONSE: While we accept this criticism, we point to the prior research that has been done into the authenticity of experiences, including restaurants, and the vast majority has been place-based, with some studies focusing on a single restaurant. We have added a cautionary note about the generalizability of the results in the limitations paragraph at the beginning of the discussion section.
COMMENT: Given that the issues of authenticity / localness / seasonality / organic ingredients arose from the data, it would have been a good idea to have included some more literature on some of these issues, or to consider more about the differences / similarities / implications of these concepts in the discussion and conclusion. For example, you could have made much more about the very interesting finding about urban and countryside restaurants that showed:“In contrast to the ‘tradition’ dimension, there were few fundamental differences between city and country restaurants in attitudes towards local and seasonal, with the primary difference being that city restaurants tended to highlight the connection to the countryside, while country restaurants tended to include the relationship to the ingredients as part of their identity.”
RESPONSE: This comment has been addressed in response to other comments that asked for more analysis, reflection, and discussion.
COMMENT: I really think much more could be made about the implications of all of your findings in the analysis / discussion and conclusions section of the manuscript. You are saying what you found, but not necessarily what are the implications of your findings, both for scholars and for practitioners.
COMMENT: covering more of the concepts to be discussed in more depth in the literature review
COMMENT: being much more explicit about the implications of the various findings in the analysis / discussion and conclusion because at the moment they tend to be more descriptive
RESPONSE: The restructuring of the results and discussion sections, along with the addition of new literature in the discussion has enabled us to increase the amount of analysis and to be more explicit about the implications.
COMMENT: Given that you have submitted this manuscript in consideration for publication in the journal Sustainability, I would have expected to have seen at least some reference to what these findings mean for restaurant or place sustainability – you have already alluded to this with regard to the carrying on of tradition, family restaurants being handed down through generations, sustainability of cultural heritage and traditions (even if it is schnitzel with fries) etc. But all of these could have been mentioned more explicitly with reference to sustainability.
RESPONSE: Some text has been added in the conclusions section to this end.
COMMENT: You conclude that “On a practical level, these results provide a basis for future research that could guide restauranteurs’ decisions when developing and implementing their restaurant concepts.” It would be useful to include a short summary paragraph of these results, showing what could / should be done for both countryside and urban restaurants to improve authenticity as something that can add value to diner experiences – especially as this reflects the title and focus of your manuscript.
RESPONSE: A brief summary of the results has been included in the conclusions section, and a figure showing the main points has been added to the discussion section. Some recommendations have been added to the conclusions section.
Reviewer 2 Report
Dear authors,
Thank you for the opportunity to review your manuscript 'The role of authenticity in adding value to experiences in non-themed Swiss restaurants'. Although I am not an expert in this topic, I did many qualitative studies myself. Furthermore, it was an interesting topic to learn more about. Thank you for that experience. I think the authors put many efforts in creating a complete overview of current science and also their additions to science are clear.
I have some suggestions for you to take into consideration:
Introduction & Authenticity of experience
When reading your introduction I really miss some information that you state in part 2 authenticity of experience. It might be a personal preference, but I suggest to make just one coherent introduction.
I can also appreciate the thorough description with examples of authenticity of experience studies you included in part 2. However, I think for a general reader this is a lot to grasp. The final two paragraphs tell more about the dimensions you took in mind when analyzing the interviews, right? You might consider including this information in the method section, like a theoretical framework.
The information that is too much for the introduction and does not fit in the methods/theoretical framework, you might consider adding in the discussion section to compare your findings more in depth with findings of previous studies.
Methods
I would consider making separate section e.g. research design, recruitment procedure, data collection procedure, theoretical framework and analysis.
Now I miss some information e.g. which questions did you ask, did you do the coding together?, how did you decide that 30 interviews were enough, how did you recruit those representatives.
By separating this information, I think it will be easier for you to be more complete in this section.
Results and discussion
Separating results and discussion might provide some more order.
The start of each section, is actually more method based information. It shows your theoretical framework.
I actually like it that you include the quotes in the text, but again try to prevent an overload of quotes... 'It is time to kill your darlings, my dear :)'
Conclusions
Limitations may fit better in the discussion section... I think all information is available in this section, but consider just to state the key lessons you learned from your interviews and their implications.
Author Response
COMMENT: When reading your introduction I really miss some information that you state in part 2 authenticity of experience. It might be a personal preference, but I suggest to make just one coherent introduction.
RESPONSE: The former ‘literature review’ section has been integrated into the introduction to make one introduction section.
COMMENT: I would consider making separate section e.g. research design, recruitment procedure, data collection procedure, theoretical framework and analysis.
RESPONSE: The text in the methodology section has been divided into subsections as suggested
COMMENT: how did you recruit those representatives?
RESPONSE: The recruitment method is now described in the subsection labeled ‘recruitment’.
COMMENT: how did you decide that 30 interviews were enough
RESPONSE: The sampling strategy is now explained in the subsection labeled ‘recruitment’.
COMMENT: which questions did you ask?
RESPONSE: The full interview list of questions has been included in the sub-section labeled ’data collection’.
COMMENT: I can also appreciate the thorough description with examples of authenticity of experience studies you included in part 2. However, I think for a general reader this is a lot to grasp. The final two paragraphs tell more about the dimensions you took in mind when analyzing the interviews, right? You might consider including this information in the method section, like a theoretical framework.
RESPONSE: This text is now included in the methodology section, which now includes a ‘theoretical framework’ subsection.
COMMENT: Did you do the coding together?
RESPONSE: The coding was done by one person, and the coding scheme then discussed within the research team to find agreement. Text has been added to describe this.
COMMENT: I actually like it that you include the quotes in the text, but again try to prevent an overload of quotes... 'It is time to kill your darlings, my dear :)'
RESPONSE: We have done our best to limit the quotes to one per paragraph.
COMMENT: The information that is too much for the introduction and does not fit in the methods/theoretical framework, you might consider adding in the discussion section to compare your findings more in depth with findings of previous studies.
RESPONSE: The introduction has been shortened and is now in one single section. The results and discussion sections have been separated, and the literature used in the discussion section
COMMENT: Limitations may fit better in the discussion section... I think all information is available in this section, but consider just to state the key lessons you learned from your interviews and their implications.
RESPONSE: The limitations have been moved to the new discussion section. The implications are also discussed in the discussion section, and the conclusions have been reduced to the key messages.
COMMENT: The start of each section, is actually more method based information. It shows your theoretical framework.
RESPONSE: This point has been addressed as part of the reorganization in response to other comments.
Round 2
Reviewer 1 Report
The authors have done well to make the suggested changes to the manuscript, and also in their explanations of what changes were made, and where suggestions could not be implemented.
I really hope you feel that my constructive suggestions have improved this article because I think you have done a really good job, and the paper now makes much more of its original contribution to knowledge than the first submission.
I also like the changed title, it much better reflects the content of the paper, and should engender interest in your work from others researching similar themes.
Author Response
Thank you! We very much appreciated the thoughtful and constructive comments in the first round of revisions, and I think we all agree that the paper has significantly improved as a result.
We are not yet finished with the topic of authenticity in domestic restaurants, and would certainly welcome it if you would choose to reach out in the future so we could exchange thoughts. I think we should be easy to find.
Reviewer 2 Report
Thank you for all the hard work and extensive adaptations. Well-done!
Author Response

(The authors gave the same response as above.)
